# The Effect of Moisture on the Adhesion Energy and Nanostructure of Asphalt-Aggregate Interface System Using Molecular Dynamics Simulation

**DOI:** 10.3390/molecules25184165

**Published:** 2020-09-11

**Authors:** Wentian Cui, Wenke Huang, Zhicheng Xiao, Jiawen Xie, Bei Hu, Xu Cai, Kuanghuai Wu

**Affiliations:** School of Civil Engineering, Guangzhou University, Guangzhou 510006, China; wentiancui1@163.com (W.C.); xzc297055783@163.com (Z.X.); xjw850843479@163.com (J.X.); 13026335960@163.com (B.H.); cx_caixu@163.com (X.C.)

**Keywords:** mineral surface, asphalt-aggregate interface, molecular dynamics, hyperspectral image technique, interfacial adhesion, moisture effect

## Abstract

In this work, the influences of moisture intruded into the asphalt-aggregate interface have been investigated at the atomistic scale. The molecular interactions of asphalt with limestone and granite were studied using molecular dynamics (MD) simulations and the mineral surface components of limestone and granite were detected using the hyperspectral image technique. Relative concentration and radial distribution function (RDF) were employed for the characterization of asphalt component aggregations on aggregates surface. Adhesion work and debonding energy were also evaluated to investigate interface energy variations in asphalt-aggregate systems. MD results showed that the presence of interfacial moisture modified asphalt nanostructure and affected the aggregation state and distribution characteristics of asphalt components near aggregate surface. The study also demonstrated that the external moisture that intruded into the interface of the asphalt-aggregate system can decrease the concentration distribution of the asphalt components with powerful polarity on aggregate surface, reduce the adhesion works of the asphalt-aggregate interface, and decline the water damage resistance of asphalt mixture.

## 1. Introduction

Asphalt mixtures are multiphase composites containing mineral powder, aggregate, and asphalt. The structure of the mixture is stabilized jointly by the support of skeletons and bonding of interfaces [1,2]. Therefore, mixture performance simultaneously depends on ingredients themselves and interfacial behaviors of various ingredients. Interfacial interaction refers to complicated physicochemical processes that take place at the interface when materials with different phases make contact [3,4]. There are a large number of interfaces in asphalt mixtures whose performances directly affect the overall performance of the mixture. Usually, interface cracks originate from microstructure defects [5,6].

Water interaction at the interface breaks the bond between asphalt and aggregate, causing moisture damage and stripping in asphalt, starting from the aggregate surface [7,8]. Nobakht et al. [9] established an adhesive and cohesive moisture damage model capable of predicting the effect of water on asphalt concrete. The results showed that the tensile bond strength of asphalt mixture was decreased due to the accumulation of water at the interface. Wang et al. [10] applied the bending beam rheometer method to investigate moisture’s effect on hot mix asphalt. Guo et al. [11] developed an asphalt mixture model and evaluated the moisture diffusion model through finite element analysis. The obtained results illustrated that pressure and temperature could accelerate moisture diffusion in asphalt mixture. Luo et al. [12] studied the diffusion of water vapor into asphalt mixture under different humidity conditions and developed two diffusion models. At the same time, obtaining the nanoscale details of interface cracking directly through experiments and the simulation of intermolecular interactions using finite element simulation technology are difficult. Since asphalt is a non-linear temperature-sensing material whose composition is complicated and changeable, studying the structural characteristics, interface failure mechanisms, and micro-scale interactions of the asphalt-aggregate interface phase is difficult.

Today, molecular dynamics (MD) simulation is being applied for the investigation of interface interactions and adhesive abilities of asphalt and aggregate. MD simulation has been proven as a powerful computational method for the simulation of the molecular interaction and interface failure of the asphalt-aggregate interface at the atomistic scale. Sun and Wang [13] studied water’s effect on the adhesive ability of virgin, aged asphalts with aggregate layer (SiO_2_ and calcite) using MD simulations. Cui et al. [14] employed mineral components such as SiO_2_ and Al_2_O_3_ as representative aggregates in a series of MD simulations. Moreover, for the simulation of moisture intrusion effect, water molecules were added into the asphalt-aggregate interface. Long et al. [15] investigated the influences of moisture and sodium chloride solution on SiO_2_ surface through simulation results. Liu et al. [16] developed a variety of asphalt-aggregate interface models and adhesion works were calculated under wet and dry conditions. Xu et al. [17] found that moisture intrusion negatively affected the adhesive ability of asphalt-aggregate interfaces. Chu et al. [18] adopted six SiO_2_ surfaces as model aggregates and adhesion work and bonding energy were investigated at the atomistic scale through MD simulations. Previous research works have investigated the effects of moisture on asphalt-aggregate interfaces and their results complied well with experimental results. Therefore, MD simulations can be applied in the investigation of the water intrusion effect at the nanoscale.

In previous works, when establishing asphalt-aggregate interface models through MD simulations, aggregates were mostly chosen from five representative oxide crystals (CaO, MgO, SiO_2_, Al_2_O_3_, and Fe_2_O_3_) [19]. However, these five oxides do not independently exist in minerals and they are mixed and interwoven with other substances under natural conditions [20,21]. Therefore, investigation of the interactions of a single oxide crystal with asphalt components using MD simulations affects the accuracy of actual molecular interactions between asphalt and aggregates. In this research, the hyperspectral image technique was applied for the detection of mineral surface components in weakly alkaline aggregate limestone and acid aggregate granite. The main minerals of limestone and granite were employed as aggregates in MD simulations, which provided more precise results for asphalt-aggregate interactions.

## 2. Results and Discussion

### 2.1. The Distribution of Asphalt Compositions on Aggregates

For further characterization of the variations of molecular structures of asphalt components near aggregates, the RDF of saturates, aromatics, resins, and asphaltenes (SARA) was extracted according to trajectories. Asphalt component RDFs were evaluated for 200 ps under constant volume and temperature (NVT) ensemble and the obtained results are shown in Figure 1a,b and Figure 2a,b.

Figure 1a shows that resin distribution reached its maximum value at 18 Å, which was higher than that of other components. Meanwhile, the concentration *g (r)* of asphaltene was greater than those of the other three components from 34.5 Å. This meant that, at asphalt-calcite interfaces, electrostatic effects between faintly alkaline minerals and polar components (asphaltene and resin) were relatively powerful. The distributions of aromatic and saturate were more uniform. Figure 1b shows that the peak values of all four components were firstly found at 2 Å from mineral surface due to the attraction of the Na^+^ ion with powerful polarity on albite surface. Then, the peak *g (r)* of weakly polar aromatic reached its peak value. Besides, the peak value and curve trend of saturate and resin were similar. These indicated that the asphalt-acidic mineral interaction was feeble and the aggregation states of four asphalt components near the surface of acidic minerals were not easily affected by minerals, and the spread of SARA was well-distributed.

As can be seen in Figure 2a, asphaltene distribution was obviously different from those of other components at the asphalt-water-calcite interface, and a tiny peak was observed at 15 Å. Due to their powerful polarizing ability, water molecules had the greatest charge effect on asphaltene, which had the strongest polarity in asphalt. This resulted in more aggregated asphaltene on the surface of weakly alkaline aggregates.

In Figure 2b, due to the moisture effect on the asphalt-water-albite interface, the peak value of *g (r)* of resin at 24 Å was decreased from 1.21 (dry condition) to 1.13, and the peak values of all curves were declined. Under wet conditions, the peak values and curve trends of saturate and resin were similar after 14 Å, and the peak value of asphaltene at 40 Å was slightly higher than that in arid condition at 40 Å, which could be due to the powerful polarity of water and asphaltene.

Concentration distribution along the vertical direction was another method which was employed for further evaluation of concentration values along different directions on aggregate surface. Dynamic trajectory was applied for determination of relative concentration along the X-direction, as illustrated in Figure 3 and Figure 4.

For the asphalt-calcite interface, the peak asphaltene value at 10–14 Å after water addition was higher than that under the dry condition. This was due to electrostatic attraction between water and asphaltene, which increased asphaltene concentration on calcite surface. The decrease of peak asphaltene value at 30 Å could be due to the presence of water, which seriously stripped asphalt film from mineral surface. For the resin with polarization capacity only lower than that of asphaltene, moisture reduced the peak concentration values of many resin peaks, which resulted in more uniform concentration distribution of resin on calcite surface. It was observed that water reduced the adhesion of minerals and asphalt by decreasing resin concentration near the surface of weakly alkaline minerals. Saturate aggregation on aggregate surface was slightly decreased and its diffusion degree was increased under the effect of moisture.

For the asphalt-albite interface, when the external free water intruded into the interface, the concentration of saturate close to the aggregate surface declined, and the asphaltene concentration was improved. The reason is that the interfacial moisture can make the non-polar light component saturate away from the mineral, attracting asphaltene on aggregate surface by the charge effect. It was found that moisture affected the appearance time and concentration of *g (r)* peaks of SARA components. In other words, although water can improve polar substances’ concentration on mineral surface, it would increase the distance and reduce the concentration distribution between asphalt and acid minerals. These result in the decrease of adhesive capacity of the asphalt-aggregate interface. The relative concentrations of SARA components illustrated that the concentrations of polar substances in asphalt on aggregate surface were gradually declined after water intruded into the asphalt-aggregate interface. At the same time, asphalt components moved away from the aggregate due to the presence of water. These phenomena were consistent with the observations of the RDF curve.

### 2.2. Adhesion Energy of the Asphalt-Aggregate System

The main reason for asphalt-aggregate system destruction was the loss of asphalt-aggregate adhesion. Therefore, the MD method was applied for the simulation of asphalt-mineral interface adhesion. In this way, interface adhesion failure behavior and the strength failure mechanism due to moisture penetration into the asphalt-aggregate interface was effectively analyzed at the microscale. This could provide researchers with reasonable technical methods to improve asphalt-aggregate system adhesion, enhance water damage resistance of asphalt mixture, and prevent water damage to asphalt pavements.

Adhesion work is defined as the energy required for the separation of interface in vacuum and is employed to describe adhesive interaction at the aggregate-asphalt interface, which reflects the bonding strength of the asphalt-aggregate system. Aggregate-asphalt interface adhesion energy was obtained by Equation (1) [22]:(1)$$W_{adhesion}=∆E_{asphalt-aggregate}=E_{total}-E_{asphalt}-E_{aggregate}$$
where, $W_{adhesion}$ is aggregate-asphalt adhesion work, $∆E_{asphalt-aggregate}$ is aggregate-asphalt interaction energy, $E_{total}$ is the total of asphalt-aggregate potential energy, and $E_{asphalt}$ and $E_{aggregate}$ are aggregate and asphalt model potential energies, respectively.

Because of the hydrophilicity of minerals and the hydrophobicity of asphalt, the water that intruded into the asphalt-aggregate interface was able to separate the two-phase interface and affect aggregate-asphalt adhesion work. Therefore, debonding work was applied to evaluate the energy required by moisture to displace asphalt from the asphalt-aggregate interface, as defined in Equation (2) [13,22]:(2)$$\;W_{debonding}=∆E_{asphalt-water}+∆E_{aggregate-water}-∆E_{asphalt-aggregate}$$
where, $W_{debonding}$ is debonding work when moisture tears asphalt from mineral surface, and $∆E_{asphalt-water}$, $∆E_{asphalt-aggregate}$, and $∆E_{aggregate-water}$ is asphalt-water, asphalt-aggregate, and aggregate-water interfacial bonding energies, respectively.

To quantify the integrated effect of adhesion work and debonding energy, energy ratio (ER) was defined as the ratio of dry to wet condition debonding energies, as presented in Equation (3) [23]:(3)$$ER=\left|W_{adhesion}/W_{debonding}\right|$$

Meanwhile, decrease percentage of adhesion work was also applied to indicate water effect on asphalt-aggregates adhesion, as expressed in Equation (4) [14]:(4)$$∆W=\left|\frac{W_{adhesion-wet}-W_{adhesion-dry}}{W_{adhesion-dry}}\right|\times 100\%$$
where, ∆W is the decreased percentage of adhesion work, and $W_{adhesion-wet}$ and $W_{adhesion-dry}$ are adhesion works under wet and dry conditions, respectively.

Asphalt-aggregate adhesion works are presented in Figure 5. Under the dry condition, the adhesion works of asphalt-calcite and asphalt-albite interfaces were found to be 467.81 and 551.01 kcal/mol respectively, which were slightly higher than that of the asphalt-calcite system. The reason for this was that, although the acidic mineral albite (Na_2_O·Al_2_O_3_·6SiO_2_) had higher SiO_2_ content, which had less contribution to adhesion work [24,25], it contained Na^+^, which had more powerful polarization ability than Ca^2+^. Due to the strong electrostatic attraction of Na^+^ toward asphaltene and resin with strong polarity in asphalt, asphalt-albite interface adhesion work was significantly increased. After moisture intruded into the asphalt-aggregate interface, asphalt-calcite adhesion work was reduced to 416.51 kcal/mol and its *∆W* value was 11.0%. However, asphalt-albite adhesion work was decreased to 444.50 kcal/mol with a *∆W* value of 19.3%. Moisture decreased the debonding abilities of aggregate and asphalt and the adhesion work of acid mineral albite was attenuated the most. The reason for this was that albite contained several silica which had powerful polarization ability toward water.

Silica tended to adsorb more water and its unsaturated forcefield could be compensated to decrease surface free energy. Therefore, moisture aggregation near albite surface continuously increased the distance between asphalt and albite, which obviously decreased their adhesion ability.

As seen in Table 1, asphalt-aggregate debonding work was negative, which indicated that the moisture intrusion into the asphalt-aggregate interface and consequently, asphalt separation from aggregate surface, was a desirable process which did not require external energy. The absolute value of debonding work represents the water damage resistance of the asphalt-aggregate interface. Asphalt-calcite interface debonding work was about twice than that of the asphalt-albite interface. Table 2 shows that the ER ratio of asphalt-calcite was 0.07, while that of asphalt-albite was 0.16. As mentioned above, higher debonding energy of the asphalt-aggregate interface system made asphalt mixture more resistant to water damage.

## 3. Models and Simulation Methods

### 3.1. Asphalt Binder Model

Asphalt binders are complicated chemical mixtures composed of hydrocarbons with different molecular weights and functional groups. Therefore, accurate determination of chemical structures and prediction of rheological and mechanical properties are impossible. Corbett [26] proposed that asphalt structure included saturates, aromatics, resins, and asphaltenes (SARA), which were more convincing to describe the chemical structure of asphalt binder. Li and Greenfield [27] developed an asphalt model with 12 components to represent asphalt binder, as shown in Figure 6. The detailed constituent parameters of asphalt binders are summarized in Table 3 [26,27].

In this study, Materials Studio (MS), an Accelrys’ commercial software, has been applied to support all model constructions and simulations. Simulation results obtained from the condensed-phase optimized molecular potentials for atomistic simulation studies (COMPASS) forcefield was consistent with actual measurements. Thus, the COMPASS forcefield was employed for describing potential energies and molecular interactions. Forcite module was employed for the optimization of the structure and energy of 12-component asphalt models. Then, the Amorphous cell tool was employed to construct an asphalt binder model using molecules. The initial density of the asphalt model was considered to be 0.1 g/cm^3^ to obtain a desirable structure with low atomic overlap. Afterwards, MD simulations were conducted with isothermal-isobaric (NPT) ensemble for 500 ps at 1 atm and 298 K with 1 fs time step. An Andersen barostat and Nose-Hoover thermostat were applied to control constant pressure and temperature, respectively. During the 500 ps NPT simulation, the density of the asphalt binder model reached a plateau after 300 ps, as shown in Figure 7. The assumed model was validated for asphalt using reports in previous literature [13,28,29].

### 3.2. Aggregate Mineral Model

Because of nearly continuous spectral information, hyperspectral images are widely applied in mineral investigations [30,31]. Therefore, hyperspectral image technology was used for the detection of the surface compositions of limestone and granite minerals [32,33]. The first step was to record hyperspectral images through a Pika XC2 hyperspectral imager, as seen in Figure 8. Then, the acquired images were imported into Environment for Visualizing Images (ENVI) platform. Hyperspectral images have huge data volumes, making data processing very complex. Therefore, the Minimum Noise Fraction (MNF) rotation tool [34] was applied to separate reliable estimations and random noises of data dimensionality in signal information. Then, the Pure Pixel Index (PPI) tool [35] was employed to extract end members in MNF images. Moreover, spectral mixing space n-dimensional visualization [36], which is an interaction method in image classification, was used for the extraction of the purest pixel spectrum. In addition, mineral standard spectral library files were selected as a contrast spectral library when determining the surface components of aggregates through spectral analyses. The spectral angle mapper (SAM) algorithm is the measurement of similarity between image spectral and reference spectral by the calculation of spectral angle between two vectors. Meanwhile, the spectral feature fittings (SFF) algorithm is the elimination of absorption features between reference spectral and each image spectral and cut-off of each original image pixel spectrum and continuum curve. The SAM and SFF algorithm were applied for the comparison of image spectrum and standard spectrum and to identify mineral components on aggregate surface.

Mineral compositions on granite and limestone surface are shown in Figure 9. In Figure 9a, the red area represents albite (68.8%), blue area indicates quartz (28.9%), and green area is biotite (2.3%). In Figure 9b, the cyan area represents calcite (73.2%), blue area is dolomite (22.8%), and red area is magnesite (4%). Due to the large proportions of calcite (CaCO3) and albite (Na_2_O·Al_2_O_3_·6SiO_2_) in limestone and granite, these minerals were selected as the representative minerals of limestone and granite, respectively. Figure 10 shows calcite and albite cells.

The unit cell of albite was cleaved to achieve a (0 0 1) surface [37] with fractional thickness of 3.0. Then, the 4 × 4 × 1 supercell structure of albite was formed. Finally, to form a three-dimensional (3D) periodic boundary module, a vacuum slab was added to extended molecular surface. The supercell structure method of calcite was similar to that of albite.

### 3.3. Asphalt-Aggregate Mineral System Model

After establishing asphalt and aggregate models, asphalt-calcite and asphalt-albite models were developed using the build layer tool, and for preventing the effects of periodic conditions, a 50 Å vacuum layer was added. Then, the most stable structure was obtained using an energy minimization function tool based on the smart descent algorithm, as shown in Figure 11a,b.

Moisture intrusion into the asphalt-aggregate interface could damage adhesion, which is the main consequence of water damage in asphalt mixtures. Therefore, asphalt-water-aggregate interface models were developed to study water’s effect on moisture damage resistance. The modeling method of these interfaces was similar to that under dry conditions. Then, 300 water molecules were added to the asphalt-aggregate interface to simulate the water intrusion effect. Optimized asphalt-water-calcite and asphalt-water-albite models are shown in Figure 12a,b.

The COMPASS forcefield was adopted for molecular simulation and an Anderson thermostat was applied to ensure the constant temperature of the structure [38,39,40]. Van der Waals (VdW) force was obtained using the atom-based summation method with a 15.5 Å cut-off distance, and electrostatic force was calculated by the Ewald summation method. The simulation was performed at 298.15 K at a constant volume and temperature (NVT) ensemble for 1000 ps, with time step of 1 fs, and dynamic trajectory data were recorded every 1000 steps.

### 3.4. Radial Distribution Function

In a given system, radial distribution function (RDF) defines the distance between the reference particle and any certain particle [41] and illustrates the aggregation of selected molecules with a certain reference molecule. Therefore, RDF was applied for the characterization of the microstructure of asphalt-aggregate due to interface water. The radial distribution function can be expressed as in Equations (5) and (6) [16]:(5)$$\rho g\left(r\right)4\pi r^{2}=dN$$
(6)$$\int_{0}^{\infty }\rho g\left(r\right)4\pi r^{2}dr=\;\int_{0}^{N}dN=N$$
where, ρ is interface density, *N* is the total number of molecules, *g (r)* is interval distance, and *r* is radial distance.

## 4. Conclusions

In the current research, mineral compositions at limestone and granite surfaces were investigated through hyperspectral image technology. Then, the MD simulation was applied to evaluate the influence of moisture that intruded into the asphalt-aggregate interface. The following simulation results and main conclusions were drawn:(1)According to hyperspectral technology and image processing results, calcite, dolomite, and magnesite were the main mineral components of limestone, where calcite accounted for 73.2%. Also, albite, quartz, and biotite were the main mineral components of granite, where albite accounted for 68.8%. Therefore, calcite and albite were used as aggregates in MD simulations.(2)Asphaltene and resin tended to accumulate near the surface of weakly alkaline aggregates due to charge interactions. Molecular interactions between bitumen and acid minerals were weak and the distribution of SARA components near the surface of acid aggregates was more uniform.(3)Moisture intrusion into the asphalt-aggregate interface could affect asphalt nanostructure and the distribution of SARA components on aggregate surface.(4)The existence of interfacial moisture declined asphalt-aggregate adhesion work. The adhesion works of limestone and granite were decreased by 11.0 and 19.3%, respectively. That is to say, the water damage resistance of asphalt mixture relied on adhesion work and debonding energy between asphalt and aggregate with the intrusion of water into the asphalt-aggregate interface system.

## Figures and Tables

**Figure 1 molecules-25-04165-f001:**
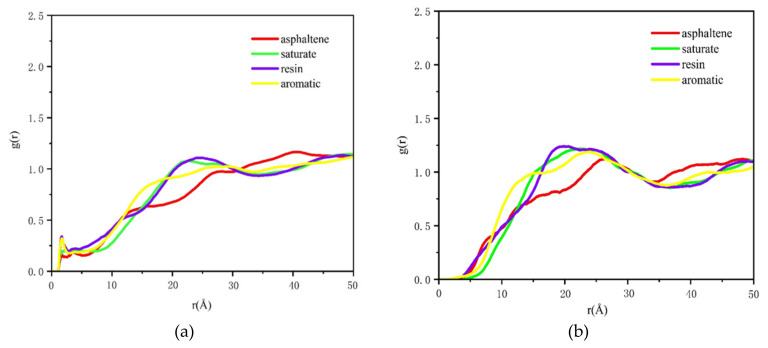
RDF curves of four asphalt components under dry condition. (**a**) Asphalt-calcite (**b**) asphalt-albite.

**Figure 2 molecules-25-04165-f002:**
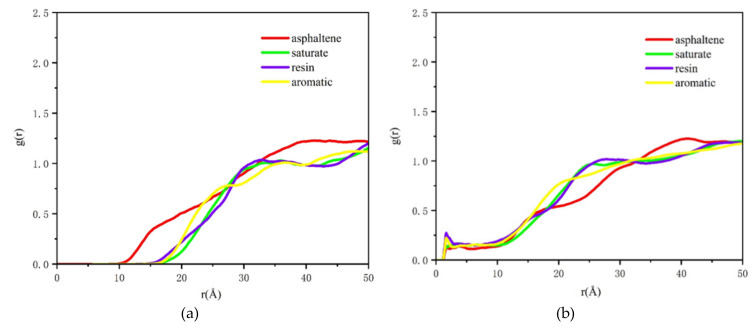
RDF curves of four asphalt components under wet condition. (**a**) Asphalt-water-calcite, (**b**) asphalt-water-albite.

**Figure 3 molecules-25-04165-f003:**
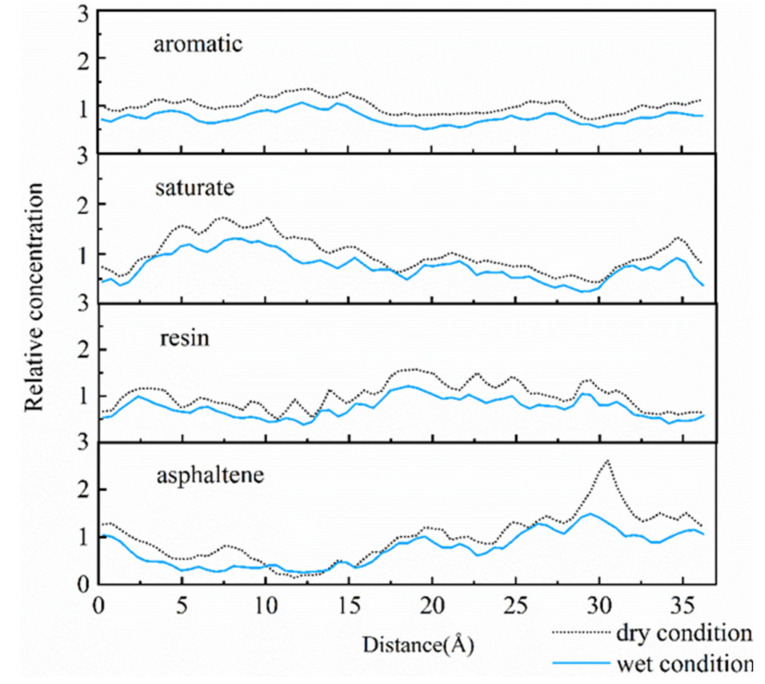
Relative concentrations of the asphalt-calcite interface system.

**Figure 4 molecules-25-04165-f004:**
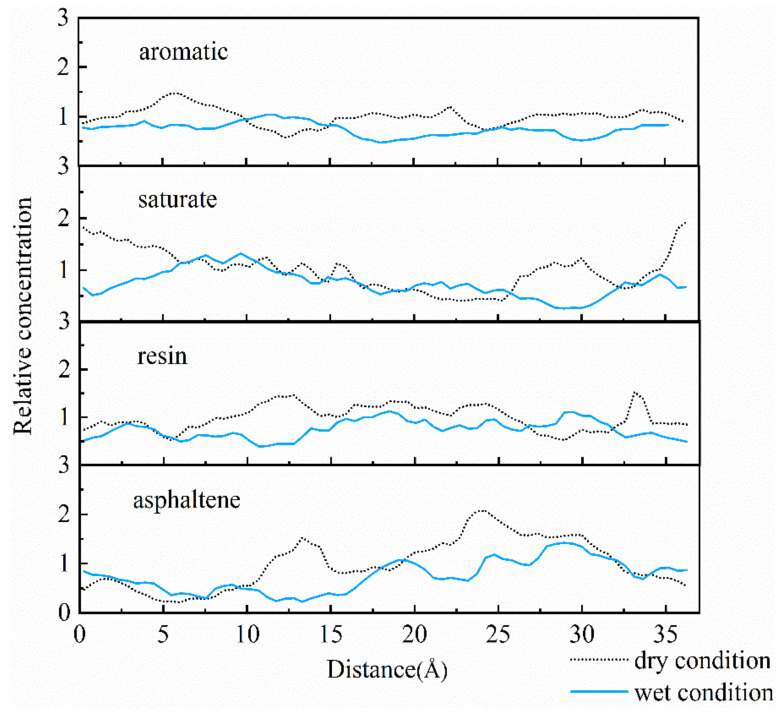
Relative concentrations of the asphalt-albite interface system.

**Figure 5 molecules-25-04165-f005:**
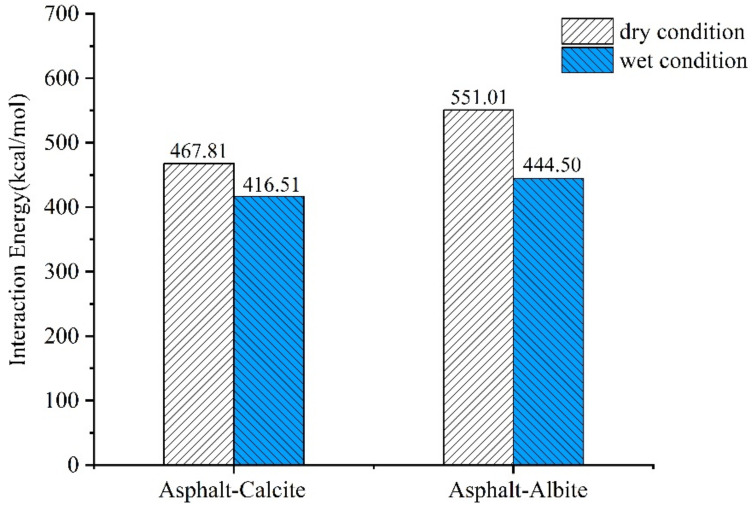
Adhesion works of the asphalt-aggregate system under different conditions.

**Figure 6 molecules-25-04165-f006:**
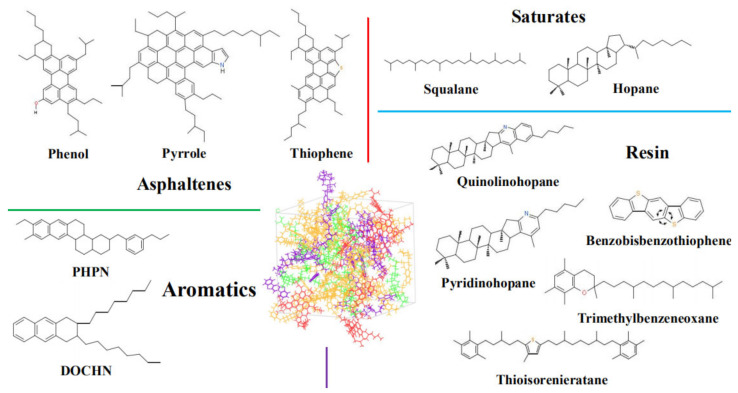
The 12-component asphalt model (asphaltene: red, resin: purple, saturate: green, aromatic: yellow).

**Figure 7 molecules-25-04165-f007:**
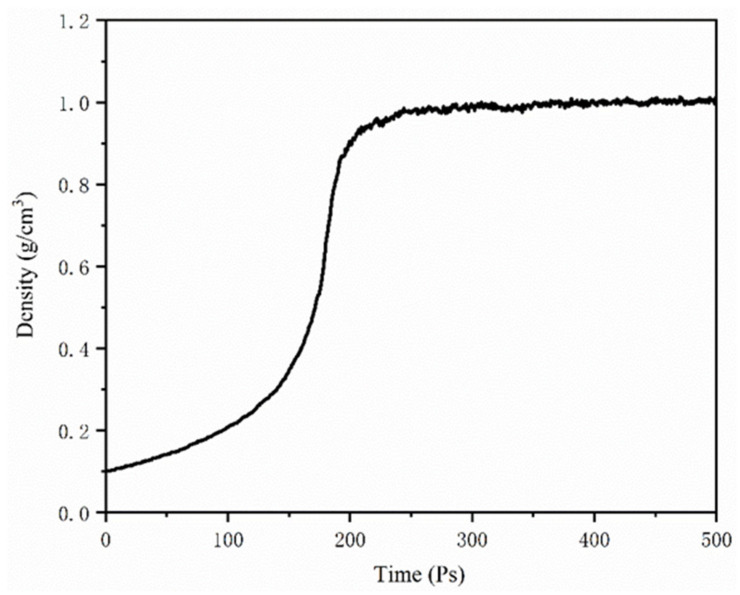
Densities of 4-component asphalt model.

**Figure 8 molecules-25-04165-f008:**
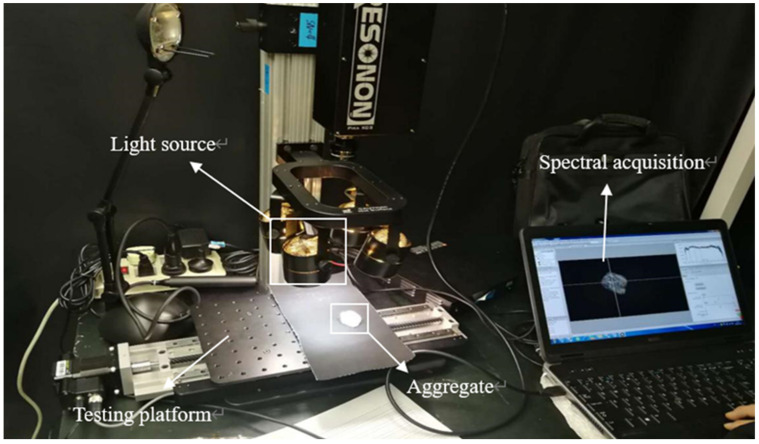
Hyperspectral test.

**Figure 9 molecules-25-04165-f009:**
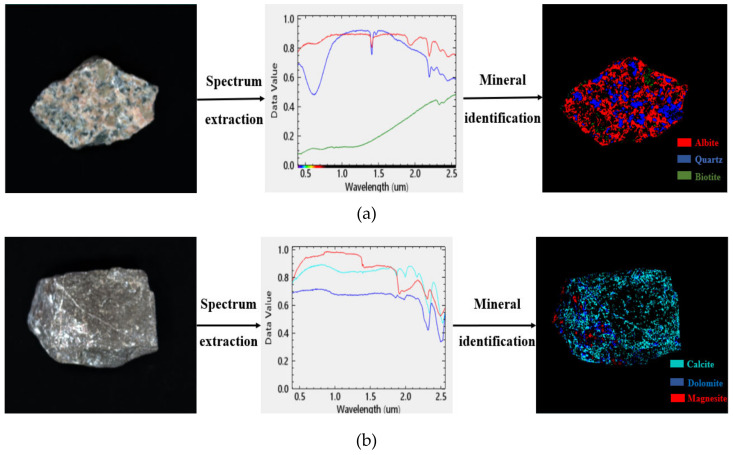
Hyperspectral identification of mineral components. (**a**) Mineral surface compositions of granite, (**b**) mineral surface compositions of limestone.

**Figure 10 molecules-25-04165-f010:**
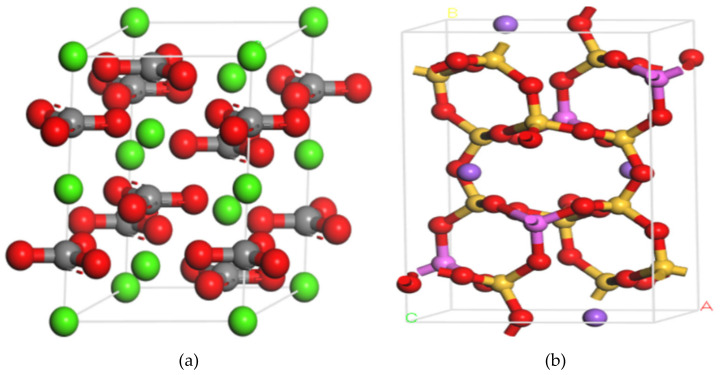
Aggregate cells of calcite and albite minerals (carbon: grey, calcium: green, oxygen: red, silica: orange, sodium: purple, aluminum: pink). (**a**) Calcite cell, (**b**) Albite cell.

**Figure 11 molecules-25-04165-f011:**
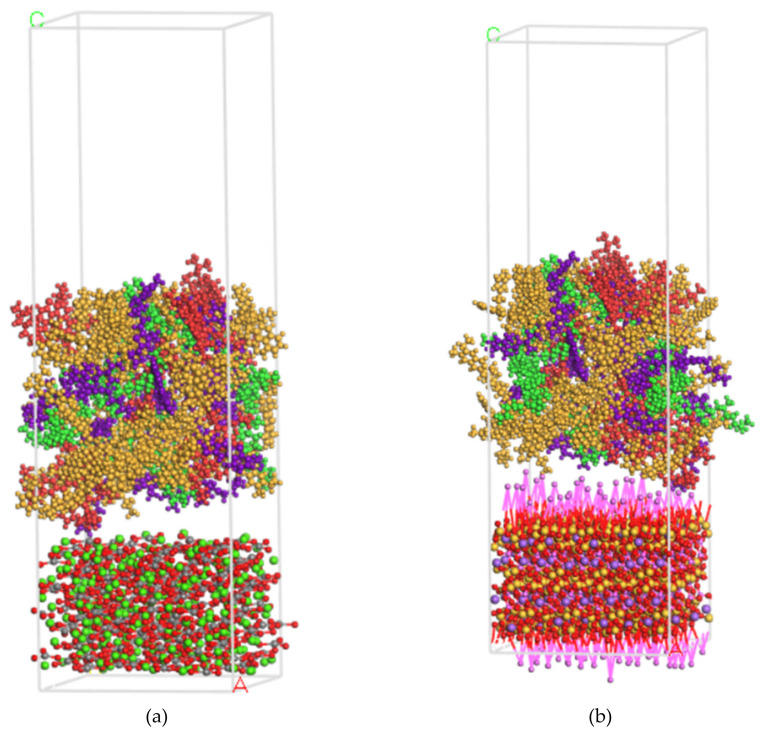
Schematic diagram of asphalt-aggregate model. (**a**) Asphalt-calcite, (**b**) asphalt-albite.

**Figure 12 molecules-25-04165-f012:**
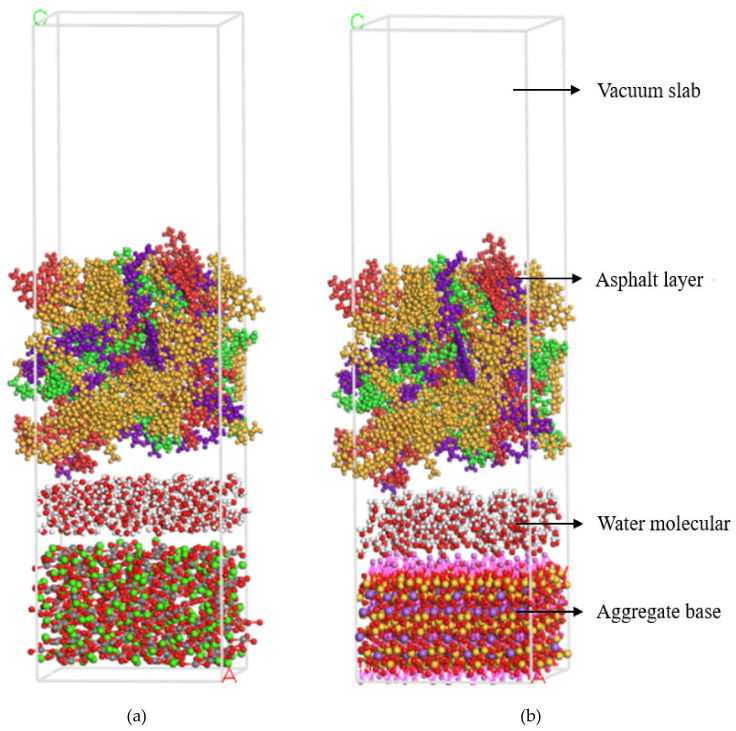
Schematic diagram of asphalt-water-aggregate model. (**a**) Asphalt-water-calcite, (**b**) asphalt-water-albite.

**Table 1 molecules-25-04165-t001:** Interface interaction and debonding energy for different models.

Models	$∆E_{asphalt-water}\;(\mathrm{kcal}/\mathrm{mol})$	$∆E_{aggregate-water}\;(\mathrm{kcal}/\mathrm{mol})$	$∆E_{asphalt-aggregate}\;(\mathrm{kcal}/\mathrm{mol})$	$W_{debonding}\;(\mathrm{kcal}/\mathrm{mol})$
Asphalt-Calcite Asphalt-Albite	7531.17 8219.80	−18,4317.15 −109,904.05	−170,136.62 −98,228.57	−6649.36 −3455.68

**Table 2 molecules-25-04165-t002:** Energy ratio (ER) calculation for different models.

Models	$W_{adhesion}(\mathrm{kcal}/\mathrm{mol})$	$W_{debonding}(\mathrm{kcal}/\mathrm{mol})$	ER (-)
Asphalt-Calcite Asphalt-Albite	467.81 551.01	−6649.36 −3455.68	0.07 0.16

**Table 3 molecules-25-04165-t003:** Detailed compositions of saturates, aromatics, resins, and asphaltenes (SARA)-component asphalt model [26,27].

Chemical Fractions	Molecules	Formula	Number	Mass Fraction (%)
Asphaltene Saturate Aromatic Resin	Phenol Pyrrole Thiophene	C_42_H_54_O C_66_H_81_N C_51_H_62_S	3 2 2	17.1
	Squalane Hopane	C_30_H_62_ C_35_H_62_	4 6	16.0
	Perhydrophe-nanthrene-naphthalene (PHPN) Dioctyl-cyclohexane-naphthalene (DOCHN)	C_35_H_44_ C_30_H_46_	13 16	43.6
	Quinolinohopane Thioisorenieratane Benzobisbenzothiophene Pyridinohopane Trimethylbenzeneoxane	C_40_H_59_N C_40_H_60_S C_18_H_10_S_2_ C_36_H_57_N C_29_H_50_O	2 2 9 2 2	23.3

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
