# Peer review of "The Effect of Moisture on the Adhesion Energy and Nanostructure of Asphalt-Aggregate Interface System Using Molecular Dynamics Simulation"

_molecules, 2020, doi:10.3390/molecules25184165_

Round 1
Reviewer 1 Report
I find the topic of this paper interesting. I have the following comments, however.
- While the text is well written, the article is not well structured. The section (#4) on modeling is placed after the section (#2) on Results, and even after Conclusions (sec. #3). Consequently, some terms introduced in sec. #4 appear already in sec. #2. This makes the paper somewhat chaotic to read.
- I find the presentation of the methods/techniques used in the paper insufficient. Their concise description should be included to enable the reader to understand the basic picture behind. At the present form the novelty and significance of the achieved results and the scientific soundness are not straightforward to grasp for a more general audience.
- In connection with point 2 above, some conclusions seem to be as if hypotheses. For example, or "electrostatic effects between ... were relatively powerful" on p. 3 or "Asphaltene and resin tend to accumulate ... due to charge interactions" on p. 8.
- The studied samples of limestone and granite are not characterized in the paper.
- Is the distance of only 2 A on p. 3 and Fig. 1b physically plausible?
- The units of quantities in several figures and Tables 2 and 3 are missing.
- The works W_adhesion and W_debonding are introduced twice on p. 6.
- The data in Fig. 5 are repeated in the text, which makes the figure redundant.
- A reference to the data and unites in Table 3 is missing. What the "Number in model"?
- I detected some typos, for example, "Figure. 10" or "4*4*1" on p. 10.
Author Response
Dear Ms. Mia Li and Reviewers,
Thanks very much for taking your time to review this manuscript. I really appreciate all your comments and suggestions! Please find my itemized responses in below and my revisions in the re-submitted files.
Sincerely,
Wentian Cui
Response to Reviewer 1 Comments:
Point 1: While the text is well written, the article is not well structured. The section (#4) on modeling is placed after the section (#2) on Results, and even after Conclusions (sec. #3). Consequently, some terms introduced in sec. #4 appear already in sec. #2. This makes the paper somewhat chaotic to read.
Response 1: Thank you for your suggestions. We are very sorry for the structure of this paper has some problems. The structure of our article followed the formatting requirements of the molecules journal: Abstract, 1. Introduction, 2. Results and discussion, 3. Materials and Methods, 4. Conclusions. Thus, the section on modeling was placed after the section on results and discussion. The conclusions position has been modified (Line338-355, page14-15). Thank you again for pointing out our shortcomings. Due to this special article structure, we introduced some terms such as energy formula description in section (#2) on results, in order to make the paper more understandable.
Point 2: I find the presentation of the methods/techniques used in the paper insufficient. Their concise description should be included to enable the reader to understand the basic picture behind. At the present form the novelty and significance of the achieved results and the scientific soundness are not straightforward to grasp for a more general audience.
Response 2: We are very appreciated with your suggestions. In the methods/techniques of this article, we elaborated on the molecular dynamics in the aspects of asphalt model, aggregate models, asphalt- aggregate interface models and force field. We have been added more information to make the hyperspectral method more substantial (Line274-280, Page10; Line292, Page11).
Point 3: In connection with point 2 above, some conclusions seem to be as if hypotheses. For example, or "electrostatic effects between ... were relatively powerful" on p. 3 or "Asphaltene and resin tend to accumulate ... due to charge interactions" on p. 8.
Response 3: Thank you for your suggestions. In this article, molecular dynamics methods are employed to establish asphalt-aggregate (albite, calcite) interface models. Then, the data of interfacial energy and concentration distributions are obtained through dynamic simulations. We analyze the data and get reliable results. However, this is an analysis of simulated results, so we adopt words such as "relatively", "tend to". etc to describe the results more objectively at atomic scale.
Point 4: The studied samples of limestone and granite are not characterized in the paper.
Response 4: Thank you for your suggestions. Owing to the complexity of actual mineral compositions of limestone and granite, we conducted hyperspectral image technology to detect the surface compositions of limestone and granite minerals. The results showed that the calcite accounts for 73.2% of limestone and albite for 68.8% of granite (Line 282-287, Page10). Because of the high proportion of calcite and albite, the characterization of calcite and albite was applied to characterize limestone and granite.
Point 5: Is the distance of only 2 A on p. 3 and Fig. 1b physically plausible?
Response 5: We are very sorry for the unclear description. 2 Å is the distance from mineral surface in asphalt-albite interface system. We have been modified (Line 90-91, Page 3). The reason for the peak is that the Na+ ions on the surface of albite attract the asphalt components.
Point 6: The units of quantities in several figures and Tables 2 and 3 are missing.
Response 6: Thank you for pointing out the missing units. We have been added (Line204, Page7; Line258, Page9). In Table 2, ER is the ratio of dry to wet condition debonding energies, it has no units.
Point 7: The works W_adhesion and W_debonding are introduced twice on p. 6.
Response 7: We are very sorry for the repetition. These have been deleted and modified (Line173,179-180, Page6).
Point 8: The data in Fig. 5 are repeated in the text, which makes the figure redundant.
Response 8: We are very appreciated with your suggestions. The data in Figure 5 and text can make the figure a litter redundant. Nevertheless, Figure 5 can intuitively show the energy change caused by the external free water intruded into the asphalt-aggregate interface. In addition, the data used in text from Figure 5 will make the sentences more fluent and convinced.
Point 9: A reference to the data and unites in Table 3 is missing. What the "Number in model"?
Response 9: We are very sorry for missing the reference and unites. The references ([26-27]) and the unites have been added (Line 241, Page8; Line258, Page9). “Number in model” in Table3 means the molecules number in asphalt model. We have been modified (Line 258, Page 9).
Point 10: I detected some typos, for example, "Figure. 10" or "4*4*1" on p. 10.
Response 10: We are very sorry for these mistakes. These have been corrected (Line282,284,287, Page10; Line289, Page11).
Reviewer 2 Report
This manuscript reports a molecular dynamics simulation study of the interaction between asphalt and two inorganic materials (limestone and granite, suitably modeled as suggested by the experimental analysis of their surface composition) both in dry and in wet conditions. The simulation study appears to be sound, and the results are interesting, so that I feel the manuscript could be published after a minor revision addressing a few points listed below.
- The last two lines of the abstract can be rephrased more clearly, I feel.
- Line 100: I cannot understand the two numerical values: from Figs. 2a and 2b I rather see half these values, namely about 0.35 and 0.25. Moreover, at line 102 there is surely some error or some missing word in the sentence " the peak value of asphaltene at 40 Å was slightly higher than that at 40 Å ".
- Lines 116-118: The decrease of the peak at 30 Å is more likely due to the fact that asphaltene approaches the calcite surface, and therefore it is less present at larger distances.
- Line 126: which peak value of the resin is meant?
- Lines 129-131: the sentence is unclear. In fact, in wet conditions the saturate is less than in the dry state very close to the surface, according to Fig. 4, but the opposite is true for asphaltene. Some clarification is required.
- Figs 3 and 4: The vertical scales for the relative concentrations should range from 0 to 2 (or 2.5), and not to 4, in order to have a better visualization of the changes between the two profiles.
- Lines 227-228: when describing the asphalt composition, what is the intended meaning of "which were more convinced"?
8. In the simulation methodology, the authors do not describe how they treated the inorganic materials in the molecular dynamics runs: did they keep it fixed, including also the sodium cations, or did they allow for their mobility?
Author Response
Dear Ms. Mia Li and Reviewers,
Thanks very much for taking your time to review this manuscript. I really appreciate all your comments and suggestions! Please find my itemized responses in below and my revisions in the re-submitted files.
Sincerely,
Wentian Cui
Response to Reviewer 2 Comments:
Point 1: The last two lines of the abstract can be rephrased more clearly, I feel.
Response 1: We are very appreciated with your suggestions. The last two lines of the abstract have been rephrased more clearly (Line19-24, Page1).
Point 2: Line 100: I cannot understand the two numerical values: from Figs. 2a and 2b I rather see half these values, namely about 0.35 and 0.25. Moreover, at line 102 there is surely some error or some missing word in the sentence " the peak value of asphaltene at 40 Å was slightly higher than that at 40 Å ".
Response 2: We are very sorry for the unclear description. The two sentences have been revised (Line101-102, Line105, Page3)
Point 3: Lines 116-118: The decrease of the peak at 30 Å is more likely due to the fact that asphaltene approaches the calcite surface, and therefore it is less present at larger distances.
Response 3: We are very appreciated with your suggestions and analysis. There are two main reasons for the decrease of asphaltene peak value. On the one hand, the 30 Å from the mineral surface is a larger distance, which results in less present asphaltene. On the other hand, the water stripped asphalt film from mineral surface can reduce the peak value of asphaltene.
Point 4: Line 126: which peak value of the resin is meant?
Response 4: We are very sorry for the unclear description. The new Figure 4 is more intuitive to show the changes of the relative concentration due to the moisture intrusion. We have made new description for Figure 4 (Line128-135, Page4) and this sentence has been deleted.
Point 5: Lines 129-131: the sentence is unclear. In fact, in wet conditions the saturate is less than in the dry state very close to the surface, according to Fig. 4, but the opposite is true for asphaltene. Some clarification is required.
Response 5: We are very appreciated with your suggestions. We have rewritten the whole paragraph (Line128-135, Page4)
Point 6: Figs 3 and 4: The vertical scales for the relative concentrations should range from 0 to 2 (or 2.5), and not to 4, in order to have a better visualization of the changes between the two profiles.
Response 6: We are very appreciated with your suggestions. The relative concentrations were ranged from 0 to 3 (Line 147 and 149, Page 5). We made analysis according to Figure 3 and Figure 4.
Point 7: Lines 227-228: when describing the asphalt composition, what is the intended meaning of "which were more convinced"?
Response 7: Thank you for your suggestion. We are sorry that we did not write the full sentence. It has been modified (Line 239, Page8).
Round 2
Reviewer 1 Report
The authors have incorporated almost all my suggestions. There are still two minor points that I recommend to be included:
- In Table 2 change "ER" to "ER (-)" to show that ER is dimensionless.
- In the caption to Table 3 include "[26-27]" at the end to emphasize the sources of the data.